

# Non-biological synthetic spike-in controls and the AMPtk software pipeline improve mycobiome data

Jonathan M. Palmer*, Michelle A. Jusino*, Mark T. Banik and Daniel L. Lindner

Center for Forest Mycology Research, Northern Research Station, USDA Forest Service, Madison, WI, USA
* These authors contributed equally to this work.

## ABSTRACT

High-throughput amplicon sequencing (HTAS) of conserved DNA regions is a powerful technique to characterize microbial communities. Recently, spike-in mock communities have been used to measure accuracy of sequencing platforms and data analysis pipelines. To assess the ability of sequencing platforms and data processing pipelines using fungal internal transcribed spacer (ITS) amplicons, we created two ITS spike-in control mock communities composed of cloned DNA in plasmids: a biological mock community, consisting of ITS sequences from fungal taxa, and a synthetic mock community (SynMock), consisting of non-biological ITS-like sequences. Using these spike-in controls we show that: (1) a non-biological synthetic control (e.g., SynMock) is the best solution for parameterizing bioinformatics pipelines, (2) pre-clustering steps for variable length amplicons are critically important, (3) a major source of bias is attributed to the initial polymerase chain reaction (PCR) and thus HTAS read abundances are typically not representative of starting values. We developed AMPtk, a versatile software solution equipped to deal with variable length amplicons and quality filter HTAS data based on spike-in controls. While we describe herein a non-biological SynMock community for ITS sequences, the concept and AMPtk software can be widely applied to any HTAS dataset to improve data quality.

Corresponding authors
Jonathan M. Palmer,
jmpalmer@fs.fed.us
Daniel L. Lindner,
dlindner@fs.fed.us

## INTRODUCTION

High-throughput amplicon sequencing (HTAS) is a powerful tool that is frequently used for examining community composition of environmental samples. HTAS has proven to be a robust and cost-effective solution due to the ability to multiplex hundreds of samples on a single next-generation sequencing (NGS) run. However, HTAS output from environmental samples requires careful interpretation and appropriate and consistent use of positive and negative controls (*Nguyen et al., 2015*). One of the major challenges in HTAS is to differentiate sequencing error versus real biological sequence variation.

Considerable progress has been made in the last several years via improved quality of sequencing results through manufacturer upgrades to reagents as well as improved quality filtering and "clustering" algorithms. While most algorithm development in HTAS is focused on the prokaryotic microbiome, using the 16S subunit of the ribosomal RNA (rRNA) array (e.g., QIIME (*Caporaso et al., 2010*), Mothur (*Schloss et al., 2009*), UPARSE (*Edgar, 2013*), DADA2 (*Callahan et al., 2016*)), many of these same tools have been adopted for use with other groups of organisms, such as fungi.

The internal transcribed spacer (ITS) region of the rRNA array has emerged as the molecular barcode for examining fungal communities in environmental samples (*Schoch et al., 2012*). The ITS region is multi-copy and thus easily amplifiable via PCR even from environmental samples with low quantities of fungal DNA. The ITS region consists of three subunits, ITS1, 5.8S, and ITS2 (Fig. 1A), and is generally conserved within fungal species yet possess enough variability to differentiate among species in many taxonomic groups. Because of its widespread use, public databases are rich with reference fungal ITS sequences (*Schoch et al., 2012*). However, there are several properties of fungi and the fungal ITS region that are potentially problematic for HTAS that include: (i) fungi have variable cell wall properties making DNA extraction efficiency unequal for different taxa and/or cell types (hyphae, fruiting bodies, spores, etc) (*Vesty et al., 2017*), (ii) the number of nuclei per cell is variable between taxa (*Roper et al., 2011*), (iii) the number of copies of the rRNA array are different between taxa and in some cases isolates of the same taxa (*Ganley & Kobayashi, 2007*), (iv) a single isolate can have multiple ITS sequences (intragenomic variability; (*Lindner & Banik, 2011*; *Simon & Weiss, 2008*)), (v) the ITS region is highly variable in length (*Schoch et al., 2014*), (vi) ITS sequences vary in GC content (*Wang et al., 2015*), and (vii) there are a variable number of homopolymer repeats (*Hart Miranda et al., 2015*). Additionally, current read lengths of commonly used sequencing platforms (Illumina Miseq currently covers ~500 bp ($2 \times 300$) and Ion Torrent is 450 bp) are not long enough to cover the entire length of the ITS region, which is typically longer than 500 bp. However, conserved priming sites exist to amplify either the ITS1 region or the ITS2 region, which has been shown to be sufficient for taxonomic identification. While several studies have used the ITS1 region for HTAS, the ITS1 region contains introns in some taxa and thus to avoid potential bias it has been suggested that ITS2 region should be the preferred region for fungi (*Taylor et al., 2016*). Progress has recently been made using single-molecule DNA sequencing (e.g., PacBio) to assess fungal communities with long read lengths (up to 3,000 base pairs), but this has not yet been widely adopted due to cost and technical hurdles (*James et al., 2016*; *Kennedy, Cline & Song, 2018*; *Tedersoo, Tooming-Klunderud & Anslan, 2018*).

Sequencing error is a known problem across NGS platforms used for HTAS. To address issues with sequencing error and reliability of results from HTAS, it has become increasingly common practice to use spiked-in "mock" community samples as positive controls for the parameterization and optimization of experimental workflows and data processing. Spike-in mock community controls for fungal ITS have been used (*Amend, Seifert & Bruns, 2010*; *De Filippis et al., 2017*; *Nguyen et al., 2015*; *Taylor et al., 2016*; *Tonge, Pashley & Gant, 2014*), and have consisted of fungal genomic DNA (gDNA)

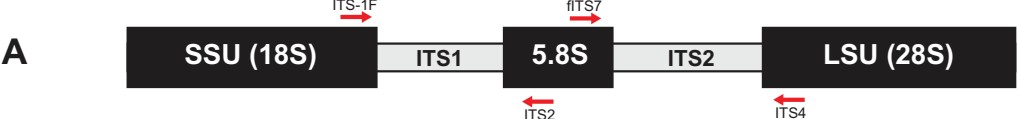

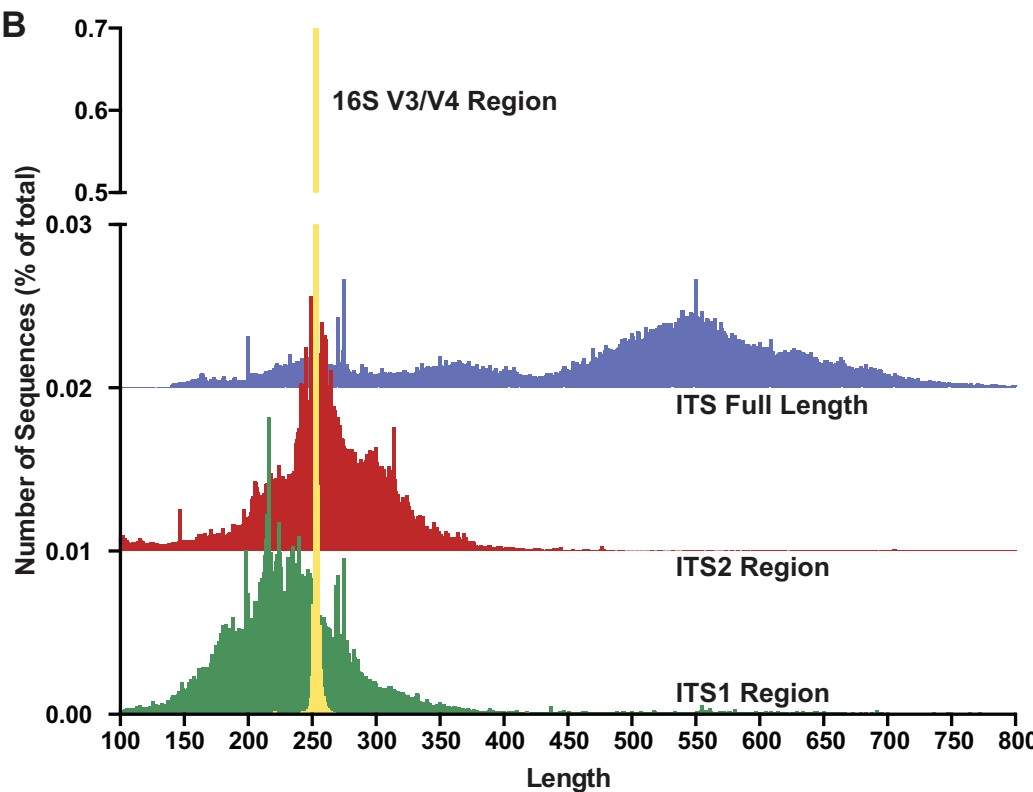

**Figure 1** **The fungal internal transcribed spacer (ITS) region of the rRNA array is highly variable in length.** (A) A schematic of the rRNA array highlights the conserved priming sites commonly used to amplify either the ITS1 or ITS2 region. (B) Size distribution of full length ITS (blue), ITS1 (green), ITS2 (red) sequences in the UNITE v7.2 curated databases shown in comparison to the bacterial 16S V3/V4 amplicon from the Silva v128 database. Current sequencing technologies do not have read lengths long enough to capture full-length ITS sequences, and thus ITS1 or ITS2 regions are used for fungal environmental community analysis. 16S V3/V4 in yellow; ITS full length in blue, ITS2 in red, and ITS1 in green. Note that the graphs for ITS2 and ITS full-length are shifted up on the y-axis and the y-axis is divided into two segments for visualization purposes.

extracted from tissue from fruiting bodies, cultures, or spores of a number of taxa which are then (usually) combined in equimolar amounts. Mock communities composed of fungal gDNA from fruiting bodies, spores, and/or hyphae provide a measure of success of extraction, PCR, and sequencing and thus are useful in the HTAS workflow. However, such mock communities are of limited value if used to validate/parameterize data processing workflows due to intrinsic properties of the ITS region mentioned previously (variable copy number, intraspecific variation, variable length, etc.). Therefore, there is a need for fungal ITS spike-in control mock communities that function to validate

laboratory experimental design, validate data processing steps, and compare results between sequencing runs and platforms.

High-throughput amplicon sequencing is cost-effective due to the ability to massively multiplex environmental samples on a single sequencing run. This process depends on the attachment of a unique sequence identifier (referred to as a barcode, an index, or a tag, depending on sequencing platform) to each piece of DNA to be sequenced. In recent years, "tag-switching" ("index hopping," "tag jumping," "barcode jumping," "index-bleed," or "barcode switching") has been noted to occur on Roche 454 platforms as well as Illumina platforms (*Carlsen et al., 2012*; *Degnan & Ochman, 2012*; *Kircher, Heyn & Kelso, 2011*; *Philippe, Lejzerowicz & Pawlowski, 2015*; *Schnell, Bohmann & Gilbert, 2015*). Tag-switching can lead to over-estimation of diversity in environmental samples (*Philippe, Lejzerowicz & Pawlowski, 2015*; *Schnell, Bohmann & Gilbert, 2015*) and mis-assignment of sequences to samples. It has been noted that spike-in mock communities may be useful to help detect tag-switching, and subsequent filters may be applied for use with the HTAS pipeline of choice (*Degnan & Ochman, 2012*; *Philippe, Lejzerowicz & Pawlowski, 2015*).

We hypothesized that a mock community composed of cloned fungal ITS sequences (in plasmids) would circumvent several of the variability issues associated with using mock communities composed of fungal DNA or fungal tissue (variable copy number, intraspecific variation, etc), allowing for a definitive assessment of HTAS for mycobiome studies. Subsequently, we found that current "off-the-shelf" software solutions performed poorly using these fungal ITS community standards and thus developed AMPtk (amplicon toolkit), a versatile software pipeline that improves results from HTAS data. Furthermore, we designed a non-biological synthetic spike-in mock community consisting of ITS-like sequences (SynMock) that, when coupled with AMPtk, provides a simple method to reduce the effects of tag-switching between multiplexed samples on a HTAS run.

# MATERIALS AND METHODS

## Biological mock community

To construct the biological mock community (BioMock) we selected 26 identified fungal cultures (Table S1) from the Center for Forest Mycology Research (CFMR) culture collection (US Forest Service, Madison, WI, USA). These cultures were purposefully chosen to represent a taxonomic range of fungal species, including species with known ITS paralogs, species with GC rich ITS regions, a variety of ITS lengths, and species with a variety of homopolymers in the ITS region. To measure the sensitivity of our bioinformatics approach, we also included two ITS sequences from *Leptoporus mollis* that were cloned from the same culture as an example of intragenomic variation in the fungal ITS region. These two sequences are more than 3% divergent (95.9% identical) and thus would typically represent separate operational taxonomic units (OTUs) in a clustering pipeline, despite being from the same fungal isolate. All cultures were grown on cellophane on malt extract agar, and DNA was extracted from pure cultures following (*Lindner & Banik, 2008*). Following extraction, the gDNA was PCR amplified using the

fungal ITS specific primers ITS-1F (*Gardes & Bruns, 1993*) and ITS4 (*White et al., 1990*). PCR products were then cloned into pGEM-T (Promega, Madison, WI, USA) and Sanger sequenced using the ITS1-F primer following the protocol in (*Lindner & Banik, 2011*). Sequences were verified via BLAST search and two clones of each isolate were selected and cultured in liquid Luria-Bertani (LB) media and incubated at 37 °C for 24 h. Plasmids were purified from the cultures in LB media using standard alkaline lysis. These plasmids will hereafter be termed "purified plasmids." The purified plasmids were then Sanger sequenced with vector primers T7 and SP6 to verify the insertion of a single copy of the appropriate ITS fragment. Purified plasmid DNA was quantified using a Qubit® 2.0 fluorometer and DNA concentrations were equilibrated to 10 nM using DNA-free molecular grade water. Following equilibration, 5 μl of each purified plasmid were combined to make an equimolar "BioMock" community of single-copy purified plasmids.

PCR has known biases, which are related to different sequence characteristics and are hard to predict in mixed DNA communities of unknown composition. To illustrate the impact of initial PCR bias on the number of reads obtained from each member of a mixed DNA community, we generated individual HTAS-compatible PCR products from each BioMock plasmid which were subsequently mixed (post-PCR) in an equimolar ratio. This was accomplished by PCR amplifying each individual plasmid with the same barcoded primer set. PCR products were purified using E-gel® CloneWell™ 0.8% SYBR® Safe agarose gels (ThermoFisher Scientific, Madison, WI, USA), quantified using a Qubit® 2.0 fluorometer, and combined into an equimolar mixture post-amplification. This post-PCR combined mock community can be used to examine sequencing error on NGS platforms and is referred to as BioMock-standards.

## Non-biological synthetic mock community

We used the well-annotated rRNA sequence from *Saccharomyces cerevisiae* as a starting point to design ITS-like synthetic sequences. The ITS adjacent regions of small subunit (SSU) and large subunit (LSU) of *S. cerevisiae* were chosen as anchoring points because of the presence of conserved priming sites ITS1/ITS1-F and ITS4. A 5.8S sequence was designed using *S. cerevisiae* as a base but nucleotides were altered so it would be compatible with several primers in the 5.8S region, including ITS2, ITS3, and fITS7. Random sequences were generated with constrained GC content and sequence length for the ITS1 and ITS2 regions. Twelve unique sequences were synthesized (Genescript, Piscataway Township, NJ, USA) and cloned into pUC57 harboring ampicillin resistance. The SynMock sequences and the script to produce them are available in the OSF repository (https://osf.io/4xd9r/) as well as packaged into AMPtk distributions (https://github.com/nextgenusfs/amptk). Each plasmid was purified by alkaline lysis, quantified, and an equimolar mixture was created as a template for HTAS library prep.

## Preparation of HTAS libraries and NGS sequencing

High-throughput amplicon sequencing libraries were generated using a proofreading polymerase, Pfx50 (ThermoFisher Scientific, Madison, WI, USA), and thermocycler conditions were as follows: initial denaturation of 94 °C for 3 min, followed by 11 cycles of

(94 °C for 30 s, 60 °C for 30 s (drop 0.5 °C per cycle), 68 °C for 1 min), then 26 cycles of (94 °C for 30 s, 55 °C for 30 s, and 68 °C for 1 min), with a final extension of 68 °C for 7 min. PCR products were cleaned using either E-gel® CloneWell™ 0.8% SYBR® Safe agarose gels (ThermoFisher Scientific, Madison, WI, USA) or Zymo Select-a-size spin columns (Zymo Research, Irvine, CA, USA). All DNA was quantified using a Qubit® 2.0 fluorometer with the high-sensitivity DNA quantification kit (ThermoFisher Scientific, Madison, WI, USA).

A single step PCR reaction was used to create Ion Torrent compatible sequencing libraries (PCR protocol described above), and primers were designed according to manufacturer's recommendations. Briefly, the forward primer was composed of the Ion A adapter sequence, followed by the Ion key signal sequence, a unique Ion Xpress Barcode sequence (10–12 bp), a single base-pair linker (A), followed by the fITS7 primer (*Ihrmark et al., 2012*). The reverse primer was composed of the Ion trP1 adapter sequence followed by the conserved ITS4 primer (*White et al., 1990*). Sequencing on the Ion Torrent PGM was done according to manufacturer's recommendations using an Ion PGM™ Hi-Q™ OT2 Kit, an Ion PGM™ Hi-Q™ Sequencing Kit, an Ion PGM™ sequencing chip (316v2 or 318v2), and raw data were processed with the Ion Torrent Suite v5.0.3 with the "–disable-all-filters" flag given to the BaseCaller. Libraries for Illumina MiSeq were generated by a two-step dual indexing strategy. All samples were PCR amplified with Illumina-fITS7 and Illumina-ITS4 primers using PCR protocol described above. PCR products were cleaned and then dual-barcoded using an eight cycle PCR reaction using the Illumina Nextera Kit and subsequently sequenced using 2 × 300 bp sequencing kit on the Illumina MiSeq at the University of Wisconsin Biotechnology Center DNA Sequencing Facility. All primers utilized in this study are available via the OSF repository (https://osf.io/4xd9r/).

## Data processing using AMPtk

AMPtk is publically available at https://github.com/nextgenusfs/amptk. All primary data and data analysis done in this manuscript are available via the Open Science Framework (https://osf.io/4xd9r/). AMPtk is written in Python and relies on several modules: edlib (*Šošic & Šikic, 2017*), biopython (*Cock et al., 2009*), biom-format (*McDonald et al., 2012*), pandas (*McKinney, 2010*), numpy (*van der Walt, Colbert & Varoquaux, 2011*), and matplotlib modules (*Hunter, 2007*). External dependencies are USEARCH v9.1.13 (*Edgar, 2010*) or greater and VSEARCH v2.2.0 (*Rognes et al., 2016*) or greater. In order to run the DADA2 (*Callahan et al., 2016*) method R is required along with the shortRead (*Morgan et al., 2009*) and DADA2 packages. The major steps for processing HTAS data are broken down into (i) pre-processing reads, (ii) clustering into OTUs, (iii) filtering OTU table, and (iv) assigning taxonomy.

### Pre-processing reads

Data structures from Roche 454 and Ion Torrent are similar where reads are in a single file and have a unique barcode at the 5′ or 3′ end of the read followed by the gene-specific priming site; therefore, AMPtk processes reads from these two platforms very similarly. As a preliminary quality control step, only reads that have a matching barcode sequence

(default is to allow 0 mismatches in barcode sequence) and forward primer are retained (default is to allow two mismatches in primer sequence). Next, reverse primer sequences are removed and if the amplicons are barcoded on the 3′ end the reverse barcode is identified. Finally, the reads are processed in a lossless trimming method that truncates reads longer than the user-defined maximum length but keeps reads shorter than this threshold if they had a valid reverse primer sequence. Data from Illumina is processed differently because reads are most often paired-end reads and most sequencing centers provide users with de-multiplexed by sample paired-end data (i.e., output of "bcl2fastq"). However, AMPtk also has de-multiplexing methods for the commonly used Earth Microbiome Project format as well as paired-end Illumina data containing barcodes sequences. When processing Illumina data, AMPtk first trims forward and reverse primer sequences, then merges the paired end reads using USEARCH or VSEARCH; phiX spike-in control is removed with USEARCH, and all data are combined into a single file. Pre-processing reads in AMPtk from any of the sequencing platforms results in a single output file that is compatible with all downstream steps.

### Clustering reads into OTUs

AMPtk is capable of running several different clustering algorithms including UPARSE, DADA2, UNOISE2, UNOISE3, and reference-based clustering. The algorithms all start with quality filtering using expected errors trimming and are modified slightly in AMPtk to build OTU tables using the de-multiplexed data (not the quality filtered data) using a 97% percent identity to OTU threshold; therefore read counts represent what was in the sample after de-multiplexing but prior to quality filtering. This is an important distinction, as using variable length amplicons coupled with quality filtering using expected error trimming (*Edgar & Flyvbjerg, 2015*) can bias longer reads, further distorting read abundance in the final OTU tables.

### Filtering OTU tables to alleviate Tag-switching

Filtering in AMPtk works optimally when a spike-in mock community is sequenced in the dataset. While by default AMPtk is setup to work with the SynMock described herein, any spike-in mock community can be used. AMPtk identifies which OTUs belong to the mock community and calculates tag-switching rate of that mock community into other samples as well as into the mock community from samples. This calculated tag-switching percentage is then used to filter the OTU table. Filtering is done on a per OTU basis, such that read counts in each OTU that are below the tag-switching threshold are set to zero as they fall within the range of data that could be attributed to tag-switching error and read counts above the threshold are not changed.

### Assigning taxonomy

AMPtk is pre-configured with databases for fungal ITS, fungal LSU, arthropod mtCO1, and prokaryotic 16S; however custom databases are easily created with the "amptk database" command. Several tools are available for taxonomy assignment in AMPtk including remote blast of the NCBI nt database, RDP Classifier (*Wang et al., 2007*), global alignment to a custom sequence database, UTAX Classifier (RC Edgar,

http://drive5.com/usearch/manual9.2/cmd_utax.html), and the SINTAX Classifier (*Edgar, 2016*). The default method for taxonomy assignment in AMPtk is a "hybrid" approach that uses classification from global alignment, UTAX, and SINTAX. The best taxonomy is then chosen as follows: (i) if global alignment percent identity is >97% then the top hit is retained, (ii) if global alignment percent identity is <97%, then the best confidence score from UTAX or SINTAX is used, (iii) if there is disagreement between taxonomy levels assigned by each method then a least common ancestor approach is utilized to generate a conservative estimate of taxonomy. AMPtk also can take a QIIME-like mapping file that can contain all the metadata associated with the HTAS study; the output is then a multi-fasta file containing taxonomy in the headers, a classic OTU table with taxonomy appended, and a BIOM file incorporating the OTU table, taxonomy, and metadata. The BIOM output of AMPtk is compatible with several downstream statistical and visualization software packages such as PhyloSeq (*McMurdie & Holmes, 2013*) and the vegan package in R (*Jari et al., 2017*).

### *Accessory scripts in AMPtk*

AMPtk has several additional features that will aid the user in analyzing HTAS data. For instance, AMPtk contains a script that will prepare data for submission to the NCBI SRA archive by formatting it properly and outputting a ready-to-submit SRA submission file. The FunGuild (*Nguyen et al., 2016*) package which assigns OTUs to an annotated database of functional guilds is also incorporated directly into AMPtk. Additionally, users can draw a heatmap of an OTU table as well as summarize taxonomy in a stacked histogram.

### HTAS software pipeline comparison

For comparison of software pipelines, sequencing data from the Ion Torrent PGM and Illumina MiSeq consisted of a single SynMock library, a single BioMock library, and 16 environmental sample libraries. To compare the outputs of these pipelines, OTUs and a corresponding OTU tables were generated using AMPtk v1.2.1, QIIME2 v2018.2, UPARSE (USEARCH v10.0.240), and PIPITS v2.0 (Illumina data only). Read length was set to 300 bp where appropriate for each tool and default or recommended settings were used for each software pipeline with the following exceptions in UPARSE: for processing the Illumina data VSEARCH was used to quality filter, de-replicate (identify unique sequences), and generate an OTU table because the file size was too large to be supported by 32-bit version of USEARCH. QIIME2 workflow utilized the "qiime cutadapt" module to de-multiplex the Ion Torrent PGM data and to remove primers from both datasets. Clustering was done in QIIME2 using DADA2 followed by 97% clustering using VSEARCH, which is the same methodology used in the "amptk dada2" method. SynMock and BioMock communities were analyzed using "amptk filter" command for each pipeline to map the OTUs to the corresponding mock communities. Error rate for each mock community was calculated as a percent of the total number of mismatches divided by the total number of base pairs. Scripts used for analysis are available at the Open Science Framework repository (https://osf.io/4xd9r/).

**Table 1 Summary statistics of the fungal ITS molecular barcode in comparison to bacterial 16S.**

| Region | Num seqs | Avg length (bp) | % HP > 6 (%) | % HP > 8 (%) | % >450 bp |
|---|---|---|---|---|---|
| ITS full length | 696,704 | 488 | 55.07 | 8.66 | – |
| ITS1 | 685,399 | 247 | 36.58 | 5.60 | 3.27% |
| ITS2 | 535,200 | 264 | 44.19 | 5.54 | 0.83% |
| 16S (V3/V4) | 627,247 | 253 | 23.74 | 1.02 | – |

**Note:**
HP, homopolymer stretches.

# RESULTS

## In silico analysis of the fungal ITS region

To gain baseline data on potential amplicons of the ITS1 or ITS2 regions, the ITS1 and ITS2 regions were extracted with the "amptk database" command using priming sites specific for each region (ITS1: ITS1-F and ITS2 primer sequences; ITS2: fITS7 and ITS4 primer sequences) from the UNITE+INSD v7.2 database (*Abarenkov et al., 2010*) consisting of 736,375 ITS sequences. For comparison, the commonly sequenced V3–V4 region was extracted using the "amptk database" command from prokaryotic 16S sequences from the Silva v128 database (*Quast et al., 2013*). A length histogram for each dataset as well as summary statistics were generated (Fig. 1B; Table 1), indicating that all three of these molecular barcodes have an average length of ∼250 bp (Table 1); however, there was considerable variation in the lengths of the ITS region in comparison to the V3/V4 region of 16S (Fig. 1B). Stretches of homopolymer sequences can also be problematic for some NGS platforms (454 and PGM), and thus the number of sequences in this dataset that contained homopolymer stretches greater than six nucleotides were calculated using the "find_homopolymers.py" script in the AMPtk distribution (Table 1). Given the small percentage of ITS1 and ITS2 regions that are greater than 450 bp (the current upper limit of the Ion Torrent PGM platform), the number of taxa in the reference database that are unlikely to sequence on the Ion Torrent due to amplicon length is relatively small (Table 1). Illumina MiSeq is now capable of paired end 300 bp read lengths (2 × 300); however, reads need to overlap for proper processing in NGS software platforms and thus a ∼500 bp size limit would also be able to sequence most taxa in the reference database using either the ITS1 or ITS2 region.

## Existing data processing workflows perform poorly with fungal ITS sequences

We cloned known ITS sequences from 26 cultures from the CFMR culture collection that varied in length (237–548 bp), ranged in GC content (43.8–68.4%), and contained sequences with homopolymer stretches with one sequence containing two 9 bp stretches. These plasmids were combined into BioMock and BioMock-standards as described in materials and methods section. The value of the BioMock-standards is that the library was combined after PCR, and thus the standards are free from PCR-induced artifacts that may arise from PCR amplification of a mixed community. Clustering amplicons into OTUs is common practice in molecular ecology and there are many software

solutions/algorithms (such as QIIME (*Caporaso et al., 2010*), UPARSE (*Edgar, 2013*), Mothur (*Schloss et al., 2009*), and DADA2 (*Callahan et al., 2016*)) that have been developed to deal appropriately with errors associated with NGS platforms. Many studies using 16S amplicon data have focused on comparing clustering methods (*Callahan et al., 2016*; *Edgar, 2013*), while others have focused on quality filtering reads prior to clustering (*Edgar & Flyvbjerg, 2015*). At the time these data were generated (2014–2015), there were very few options for pre-processing data from the Ion Torrent platform; thus we used the compatible 454 method in QIIME v1.9 to de-multiplex the sequencing data (minimum length = 100, maximum length = 550, maximum length homopolymers = 10, maximum primer mismatch = 2). These data were then clustered using UCLUST, SWARM, and USEARCH in QIIME v1.9 as well as UPARSE (usearch v7.0.1090). However the number of OTUs was highly over-estimated for our mock communities and the error rates were very high (Table S2). The best performing clustering pipeline was UPARSE; however the number of OTUs predicted for the 12 member SynMock was unacceptably high, resulting in 82 OTUs with the Ion Torrent PGM data while the 38 OTUs for Illumina MiSeq data were slightly more accurate (Table S2). Based on advances in the UPARSE pipeline and the importance of quality trimming outlined by *Edgar & Flyvbjerg (2015)*, we speculated that these high error rates and inflated OTU counts could be due to the pre-processing of reads (finding barcodes/primers and quality trimming the sequence data) allowing errors in the data. We were unable to run our data through Mothur due to the inability to do a multiple sequence alignment and subsequent distance matrix of the ITS region. It is important to note that with the exception of USEARCH/UPARSE, these software solutions have been built with 16S amplicons in mind and several have been optimized for Illumina data.

One major difference in 16S amplicons versus those of ITS1/ITS2 is that the lengths of 16S amplicons are nearly identical, while ITS1/ITS2 amplicons vary in length (Fig. 1B). The variable length coupled with sequence divergence of the ITS region between diverse taxa, makes the ITS region impossible to align in many cases (*Schoch et al., 2012*) and thus represents a major limitation in data processing (i.e., sequence alignment is required for default Mothur OTU clustering). To illustrate the importance of properly pre-processing ITS data, we clustered the ITS1 and ITS2 regions using UPARSE while using the full length ITS1/ITS2 UNITE reference database as a benchmark (Fig. 2). Using the UNITE database, we then explored the outcome of trimming/truncating the sequences to different length thresholds, a common practice in OTU clustering pipelines. The UPARSE algorithm uses global alignment and as such terminal mismatches count in the alignment (as opposed to local alignment where terminal mismatches are ignored); thus the UPARSE pipeline expects that reads are truncated to a set length. The original UPARSE algorithm achieves this by truncating all reads to a set length threshold and discards reads that are shorter than the length threshold. Therefore real ITS sequences are discarded (Fig. 2). We then came up with two potential solutions to fix this unintended outcome: (i) truncate reads that were longer than the threshold and keep all shorter reads (full length), and (ii) truncate longer reads and pad the shorter reads with N's out to the length threshold (padding). Using the UNITE v7.2 database of curated

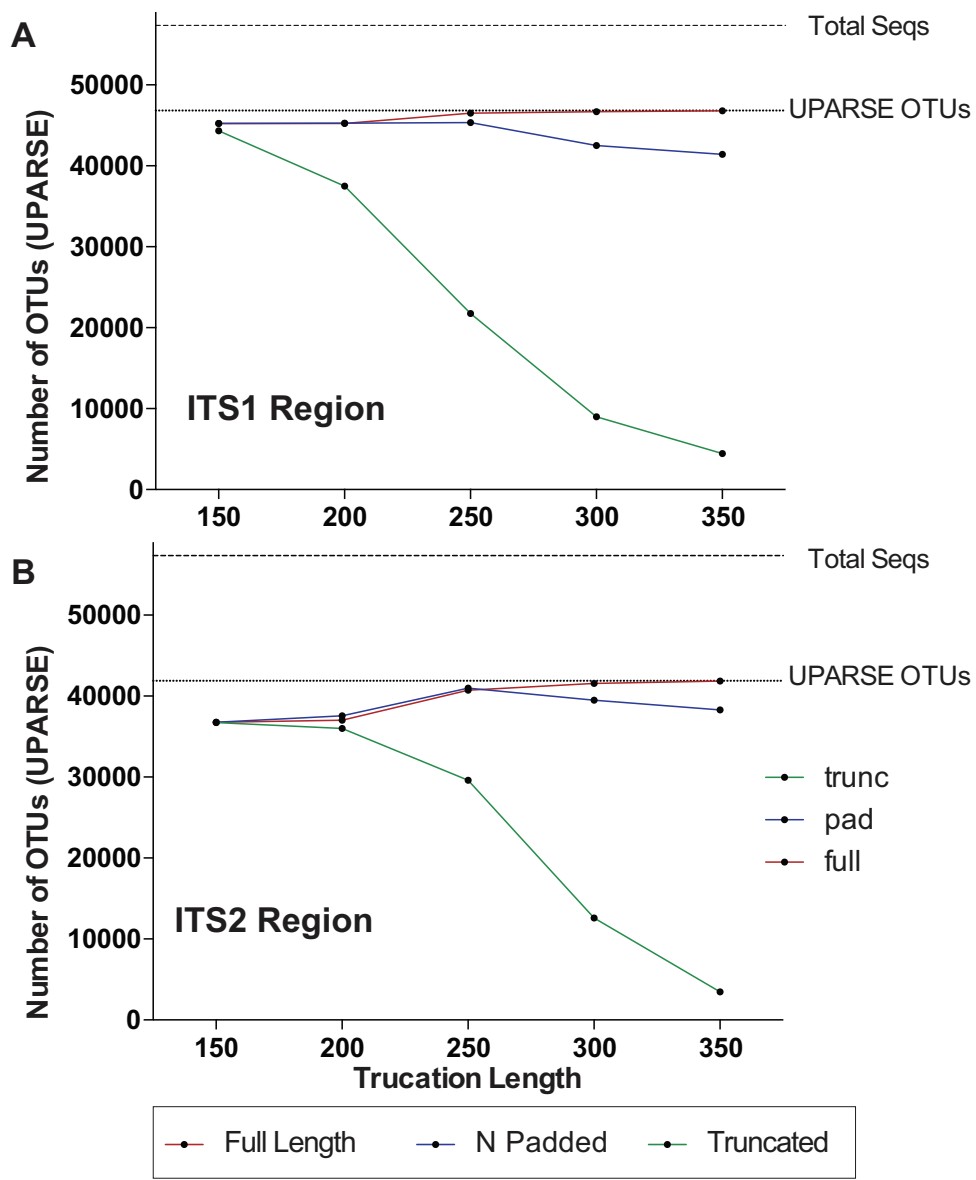

**Figure 2 Pre-processing ITS sequences is critically important to accurately recover OTUs using the curated UNITE v7.2 reference database.** ITS1 and ITS2 sequences were extracted from the UNITE v7.2 general fasta release database using "AMPtk database." Identical sequences were collapsed (dereplication) and remaining sequences were clustering using UPARSE ("cluster_otus") to generate the total number of UPARSE OTUs expected for the (A) ITS1 and (B) ITS2 regions. The data was then processed to five different lengths (150, 200, 250, 300, and 350 bp) and then clustered (UPARSE "cluster_otus") using (i) default UPARSE truncation (longer sequences are truncated and shorter sequences are discarded), (ii) padding with ambiguous bases (longer sequences truncated and shorter sequences padded with N's to length threshold), and (iii) full-length sequences (longer sequences are truncated and shorter sequences are retained if reverse primer is found). Full-length and padding pre-processing sequences outperforms default UPARSE truncation.

sequences (general release June 28th, 2017) as input, both "full-length" and "padding" improved UPARSE results with the "full length" method recovering more than 99% of the expected OTUs (Fig. 2). It should be noted that the recent version of USEARCH

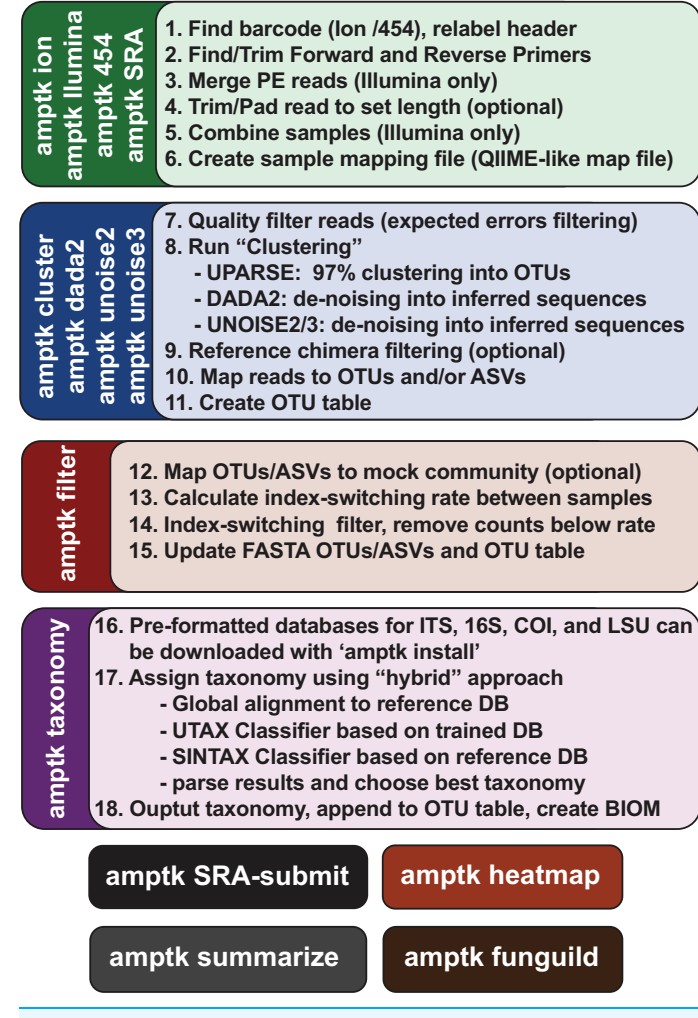

**Figure 3 Overview of the commands in AMPtk.** AMPtk is built to be compatible with multiple sequencing platforms as well as contains several clustering algorithms.

(versions >v8.0.1611) includes a similar method to allow reads to be padded with N's out to a user-defined length threshold (https://www.drive5.com/usearch/manual/cmd_fastx_truncate.html).

Due to the intrinsic nature of the variable length ITS amplicons, we needed a data processing solution that would be flexible enough to maintain the full length of the reads, trim reads without data loss, prepare sequencing reads for downstream clustering algorithms, handle a large number of sequences, and support all major NGS platforms. Using the mock communities described herein as a means to validate the results of all data processing steps, we wrote a flexible series of scripts for processing Illumina, Ion Torrent, as well as Roche 454 data that are packaged into AMPtk (amplicon tool kit). A flow diagram of AMPtk is illustrated in Fig. 3 and a more thorough description of AMPtk is provided in the material and methods section. A manual for AMPtk is available at http://amptk.readthedocs.io/en/latest/. After data are pre-processed with AMPtk via a platform specific method, AMPtk then functions as a wrapper for several popular

**Table 2 Comparison of HTAS software pipelines on the Ion Torrent PGM sequencing platform.**

| Pipeline | Total OTUs | Reads in OTU table | SynMock OTUs ($n = 12$) | SynMock error rate[1] (%) | BioMock OTUs ($n = 23$) | BioMock error rate[1] (%) | Primer/adapter contam.[2] (%) | Run time (min)[3] |
|---|---|---|---|---|---|---|---|---|
| AMPtk (UPARSE) | 901 | 1,830,315 | 30 (12) | 0.099 | 30 (23) | 0.057 | 0.00 | 17 |
| USEARCH (UPARSE) | 821 | 1,823,711 | 32 (11) | 1.573 | 32 (23) | 0.054 | 63.63 | 215 |
| AMPtk (DADA2) | 814 | 1,822,013 | 30 (12) | 0.099 | 31 (23) | 0.058 | 0.00 | 42 |
| QIIME2 (DADA2) | 277 | 676,282 | 8 (5) | 2.276 | 21 (19) | 0.043 | 97.11 | 41 |

**Notes:**
[1] Error rate is the percent of total mismatches/total number of nucleotides.
[2] Primers and adapter contamination was measured by using the "search_oligodb" method in USEARCH v10.0.240.
[3] Data were run using 10 cpus/threads on a Mac Pro OS 10.13.3 (12 core 2.4 GHz Intex Xeon processor with 64 GB of RAM).

algorithms including UPARSE, DADA2, UNOISE2, and UNOISE3. Data presented in Fig. 2 were generated using AMPtk v1.0.1 while the rest of the data were generated using AMPtk v1.2.1.

## AMPtk is a fast and accurate HTAS pipeline

Molecular ecology is a rapidly changing field because methods must keep pace with technological advances in DNA sequencing. Therefore, we compared AMPtk to a handful of currently popular HTAS software pipelines: QIIME2 (https://qiime2.org), USEARCH (*Edgar, 2013*), and PIPITS v2.0 (*Gweon Hyun et al., 2015*). PIPITS is not a universal HTAS pipeline as it only works with paired-end Illumina data; however, we included it because it uses the ITSx software (*Bengtsson-Palme et al., 2013*) to extract the ITS region. For analysis of the Ion Torrent PGM data AMPtk performed the best by recovering all members of the SynMock and BioMock communities, combined with the lowest error rate, the fastest run time (17 min), and the most read counts recovered in the OTU table (Table 2). Both USEARCH and QIIME2 resulted in OTUs that contained either primer or adapter sequence contamination, largely due to incomplete removal of the reverse primer sequence (Table 2). A similar trend was found in processing the Illumina MiSeq data; AMPtk recovered all members of the SynMock and BioMock communities, had low error rates, and recovered the most read counts in the OTU table (Table 3). USEARCH was slightly faster than AMPtk (55 vs 67 min, respectively), however, it had high error rates for the SynMock community (1.047%) as well as missed a member of the SynMock community (Table 3). The mock community members that were missed by USEARCH or QIIME2 were those that were shorter amplicon sequences, i.e., both pipelines missed mock6 (ITS2 region is 161 bp). All pipelines produced more OTUs in each mock community sample than were expected; some of these OTUs are a result of tag-switching and a few others are chimeras that slip through the chimera detection algorithms in UPARSE or DADA2. These OTUs can be filtered out of the dataset using the "amptk filter" command, however, for transparency and to measure the pipeline performance, the raw data are presented (Tables 2 and 3). PIPITS was unable to identify any members of the SynMock community because the non-biological SynMock sequences are not identified by ITSx and thus the SynMock cannot be processed by pipelines that use ITSx as a method to pull out the ITS1 or ITS2 regions (Table 3).

**Table 3 Comparison of HTAS software pipelines on the Illumina MiSeq sequencing platform (2 × 300 bp).**

| Pipeline | Total OTUs | Reads in OTU table | SynMock OTUs (n = 12) | SynMock error rate[1] (%) | BioMock OTUs (n = 23) | BioMock error rate[1] (%) | Primer/adapter contam.[2] (%) | Run time (min)[3] |
|---|---|---|---|---|---|---|---|---|
| AMPtk (UPARSE) | 2,007 | 13,636,429 | 33 (12) | 0.063 | 36 (23) | 0.031 | 0.00 | 67 |
| USEARCH (UPARSE[4]) | 1,924 | 12,073,312 | 43 (11) | 1.047 | 33 (23) | 0.029 | 0.00 | 55 |
| AMPtk (DADA2) | 1,954 | 13,628,802 | 34 (12) | 0.063 | 42 (23) | 0.032 | 0.00 | 470 |
| QIIME2 (DADA2) | 786 | 8,522,714 | 16 (10) | 1.186 | 29 (22) | 0.901 | 53.43 | 269 |
| PIPITS (ITSx) | 2,784 | 9,798,280 | 20 (0) | 100 | 48 (22) | 0.744 | 0.00 | 1,108 |

**Notes:**
[1] Error rate is the percent of total mismatches/total number of nucleotides.
[2] Primers and adapter contamination was measured by using the "search_oligodb" method in USEARCH v10.0.240.
[3] Data were run using 10 cpus/threads on a Mac Pro OS 10.13.3 (12 core 2.4 GHz Intex Xeon processor with 64 GB of RAM).
[4] VSEARCH was used for dereplication (finding unique sequences), quality filtering, and constructing OTU table due to data being too large for 32-bit USEARCH10.

## Read abundances do not represent community abundances: PCR introduces bias

Next-generation sequencing platforms are quantitative if the library to be sequenced is unbiased, as is typically the case with RNA-sequencing and whole genome sequencing library prep protocols. However, PCR of mixed communities has long been shown to introduce bias in NGS workflows (*Aird et al., 2011*; *Kebschull & Zador, 2015*; *Pinto & Raskin, 2012*) and use of 16S mock communities resulted in read abundance bias (*Bokulich et al., 2013*; *Edgar, 2017*; *Kozich et al., 2013*). For HTAS this is an important caveat, as molecular ecologists are interested in diversity metrics of environmental communities as well as their relative abundance. Through the use of mock communities, several studies have pointed out that read abundance from fungal HTAS are not representative of relative biological abundance (*Amend, Seifert & Bruns, 2010*; *De Filippis et al., 2017*). However, it was recently reported that for a fungal ITS mock community of eight members, abundances were meaningful (*Taylor et al., 2016*) and correlations between template abundance and read abundance have been previously reported (*Ihrmark et al., 2012*). Due to disagreements in the literature many studies use abundance-based metrics to analyze HTAS, perhaps due to the unintended consequences of using presence/absence metrics on data that suffer from tag-switching. We reasoned we could investigate this issue using the ITS BioMock artificial community, which would not suffer from bias associated with DNA extraction, ITS copy numbers, and intraspecific variation. We compared the relative read abundances of BioMock-standards to three different combinations of BioMock on both the Ion Torrent PGM and Illumina MiSeq platforms (Fig. 4). The BioMock-standards consist of an equimolar mixture of 26 PCR products thereby removing the PCR bias from mixed DNA samples, while the BioMock communities consist of an equimolar mixture of 23 single-copy plasmids. These data show that even in an extreme example of an equally mixed community of cloned ITS sequences, read abundance does not represent actual abundance in the mock community (Fig. 4). The majority of the bias is introduced at the initial PCR step, as the post-PCR combined BioMock-standards result in a more equal distribution of reads, albeit not a perfect distribution. We also tested PCR conditions, DNA concentrations, and sample

| Species | ITS2 Length | GC Content | HP > 5 | ID | Ion Torrent PGM | | | | Illumina MiSeq | | | |
|---|---|---|---|---|---|---|---|---|---|---|---|---|
| | | | | | Stds | Mock A | Mock B1 | Mock B2 | Stds | Mock A | Mock B1 | Mock B2 |
| *Phialocephala fusca* | 237 | 68.4% | 0 | mock1 | 4905 | 19 | 6 | 1 | 8615 | 725 | 329 | 3337 |
| *Ascomycete sp.* | 238 | 50.8% | 0 | mock2 | 5106 | 11651 | 10809 | 11877 | 9174 | 20763 | 26129 | 18341 |
| *Phialocephala lagerbergii* | 238 | 58.8% | 0 | mock3 | 4886 | 13479 | 12111 | 13392 | 8648 | 28515 | 29482 | 21269 |
| *Helotiales sp.* | 239 | 57.3% | 0 | mock4 | 4233 | 15219 | 13048 | 14896 | 9050 | 27726 | 32576 | 24276 |
| *Aspergillus candidus* | 260 | 65.8% | 3 | mock5 | 2813 | 31 | 23 | 3 | 8992 | 147 | 122 | 269 |
| *Bjerkandera adusta* | 281 | 51.2% | 0 | mock6 | 3977 | 8112 | 7172 | 7787 | 13597 | 13112 | 13866 | 15067 |
| *Laetiporus caribensis* | 283 | 52.7% | 0 | mock7 | 3330 | 7810 | 6457 | 6365 | 9404 | 15035 | 16622 | 16385 |
| *Trametes gibbosa* | 288 | 50.0% | 1 | mock8 | 3637 | 7281 | 6914 | 6865 | 8137 | 13819 | 14579 | 14787 |
| *Laetiporus gilbertsonii* | 290 | 54.1% | 0 | mock9 | 4066 | 8831 | 10401 | 12638 | 8751 | 22860 | 21680 | 20682 |
| *Gloeoporus pannocinctus* | 292 | 43.8% | 0 | mock10 | 2603 | 2922 | 3025 | 2567 | 9718 | 11150 | 11792 | 14265 |
| *Wolfiporia dilatohypha* | 293 | 54.6% | 0 | mock11 | 3957 | 94 | 110 | 109 | 8775 | 243 | 224 | 194 |
| *Schizopora sp.* | 293 | 48.1% | 0 | mock12 | 4037 | 6965 | 7030 | 6626 | 8676 | 12857 | 13947 | 14860 |
| *Fomitopsis ochracea* | 295 | 44.1% | 0 | mock13 | 3689 | 2913 | 2860 | 2651 | 9471 | 5522 | 5432 | 6883 |
| *Laetiporus cremeioporus* | 296 | 54.7% | 0 | mock14 | 3922 | 10279 | 11920 | 12440 | 8262 | 16454 | 16390 | 16798 |
| *Phanerochaete laevis* | 300 | 47.7% | 1 | mock15 | 3863 | 6970 | 7650 | 6876 | 9242 | 15667 | 15543 | 18168 |
| *Laetiporus cincinnatus* | 302 | 54.0% | 0 | mock16 | 3133 | 5699 | 7645 | 7505 | 7675 | 16819 | 16157 | 14608 |
| *Punctularia strigosozonata* | 303 | 53.1% | 0 | mock17 | 4019 | 8271 | 7688 | 8217 | 7669 | 10701 | 11572 | 11671 |
| *Phellinus cinereus* | 314 | 49.7% | 0 | mock18 | 3672 | 2937 | 2985 | 2597 | 9807 | 6314 | 5953 | 7496 |
| *Antrodiella semisupina* | 315 | 43.8% | 1 | mock19 | 3089 | 3047 | 3406 | 2741 | 9297 | 9356 | 8990 | 11593 |
| *Leptoporus mollis* | 315 | 45.4% | 3 | mock20 | 3551 | 4969 | 4320 | 4028 | 9047 | 8847 | 8747 | 9987 |
| *Leptoporus mollis 2* | 315 | 45.1% | 1 | mock21 | 3776 | 207 | 366 | 249 | 9250 | 405 | 302 | 414 |
| *Mortierellales sp.* | 353 | 45.0% | 0 | mock22 | 3264 | 4668 | 4311 | 3812 | 9151 | 10865 | 9728 | 13365 |
| *Laetiporus persicinus* | 379 | 51.2% | 2 | mock23 | 2147 | 2651 | 2385 | 2053 | 6486 | 488 | 421 | 521 |
| *Penicillium nothofagi* | 260 | 66.2% | 1 | mock24 | 3644 | NA | NA | NA | 8278 | NA | NA | NA |
| *Metapochonia suchlasporia* | 291 | 64.6% | 1 | mock25 | 1976 | NA | NA | NA | 2045 | NA | NA | NA |
| *Wolfiporia cocos* | 548 | 59.7% | 0 | mock26 | 7 | NA | NA | NA | 5979 | NA | NA | NA |

**Figure 4 Read abundance is an unreliable proxy for actual abundance within a mixed community.** Using an equimolar mixture of cloned ITS sequences in plasmid form (MockA, MockB1, MockB2) in comparison to equimolar mixture of individual PCR products (Stds) illustrates that the initial PCR reaction during library preparation heavily biases the read abundance obtained after sequencing on both the Ion Torrent PGM and Illumina MiSeq platforms. While read abundances are unreliable, all members of the mock community were recovered. MockA represents a 1:16,000 dilution and MockB1/MockB2 are replicates of a 1:32,000 dilution of the BioMock community. The Ion Torrent PGM platform has a length threshold of approximately 450 bp; therefore longer amplicons like *Wolfiporia cocos* ITS2 sequence very poorly.

reproducibility on the Ion Torrent PGM and in all conditions tested we still observed unequal read abundances (Fig. S1).

While the bias via PCR is consistent between sequencing platforms, there is no obvious correlation between length of the read, GC content, nor stretches of homopolymers affecting efficient PCR amplification. For example, *Wolfiporia dilatophya* (mock11) contains no homopolymer stretches larger than 5, has GC distribution of 54.6%, and is near the median in length, yet it does not PCR amplify well in the BioMock community (Fig. 4). These data also show a size limitation in the Ion Torrent PGM workflow, as *Wolfiporia cocos* (mock26) sequences very poorly due to its long ITS2 region of 548 bp (Fig. 4). Three members of the original 26 members of the BioMock community were dropped (mock24, mock25, mock26) due to persistent problems getting them to amplify/sequence in repeated HTAS on the Ion Torrent platform (Fig. S1). While mock26 (*W. cocos*) was likely difficult to sequence due to its long size, mock24 and mock25 consistently failed to amplify in the equal mixture of plasmids for unknown reasons.

In HTAS experiments, considerable effort is made to try to sequence to an equal depth for each sample. However, in practice this rarely works perfectly and thus a typical HTAS dataset has a 2–4× range in number of reads per sample. The depth of sequence range for the HTAS runs presented here is within a range of 2× for each run and the smallest number of reads per sample in any of our sequencing runs was nearly 60,000 (Table S5). Unequal sequencing depth has been used as rationale for explaining the lack of correlation between read abundance and actual abundance. Therefore, random subsampling of reads in each sample prior to clustering (also called rarefying) has been widely used in the literature, despite a compelling statistical argument that this method is flawed (*McMurdie & Holmes, 2014*). Randomly subsampling reads for each sample using our BioMock community yielded nearly identical read abundance biases (Fig. S2). Sequencing depth has been shown to be an important variable for HTAS experiments (*Smith & Peay, 2014*), therefore it has been recommended that a reads per sample cutoff be used with processing environmental datasets (while each dataset is different and should be tested empirically, typically we use 5,000–10,000 as a minimum number of reads per sample).

## A non-biological synthetic mock community to measure tag-switching among samples

Tag-switching is a phenomenon that has been described on Roche 454 platform (*Carlsen et al., 2012*) as well as Illumina platforms (*Kircher, Sawyer & Meyer, 2012*; *Wright & Vetsigian, 2016*). A consensus on a mechanism of tag-switching during the sequencing run has yet to be reached. Tag-switching is a significant challenge to overcome as sample crossover has the potential to over-estimate diversity and lead to inaccurate representations of microbial communities, especially considering that read abundance is an unreliable proxy for biological abundance (Fig. 4). Using our BioMock sequencing results, we also discovered this phenomenon on both Ion Torrent and Illumina platforms. We calculated the rate of tag-switching in our BioMock Ion Torrent sequencing run to be 0.033% and on Illumina MiSeq between 0.233% and 0.264%. We also confirmed that tag-switching was happening on the Illumina flow-cell by re-sequencing a subset of

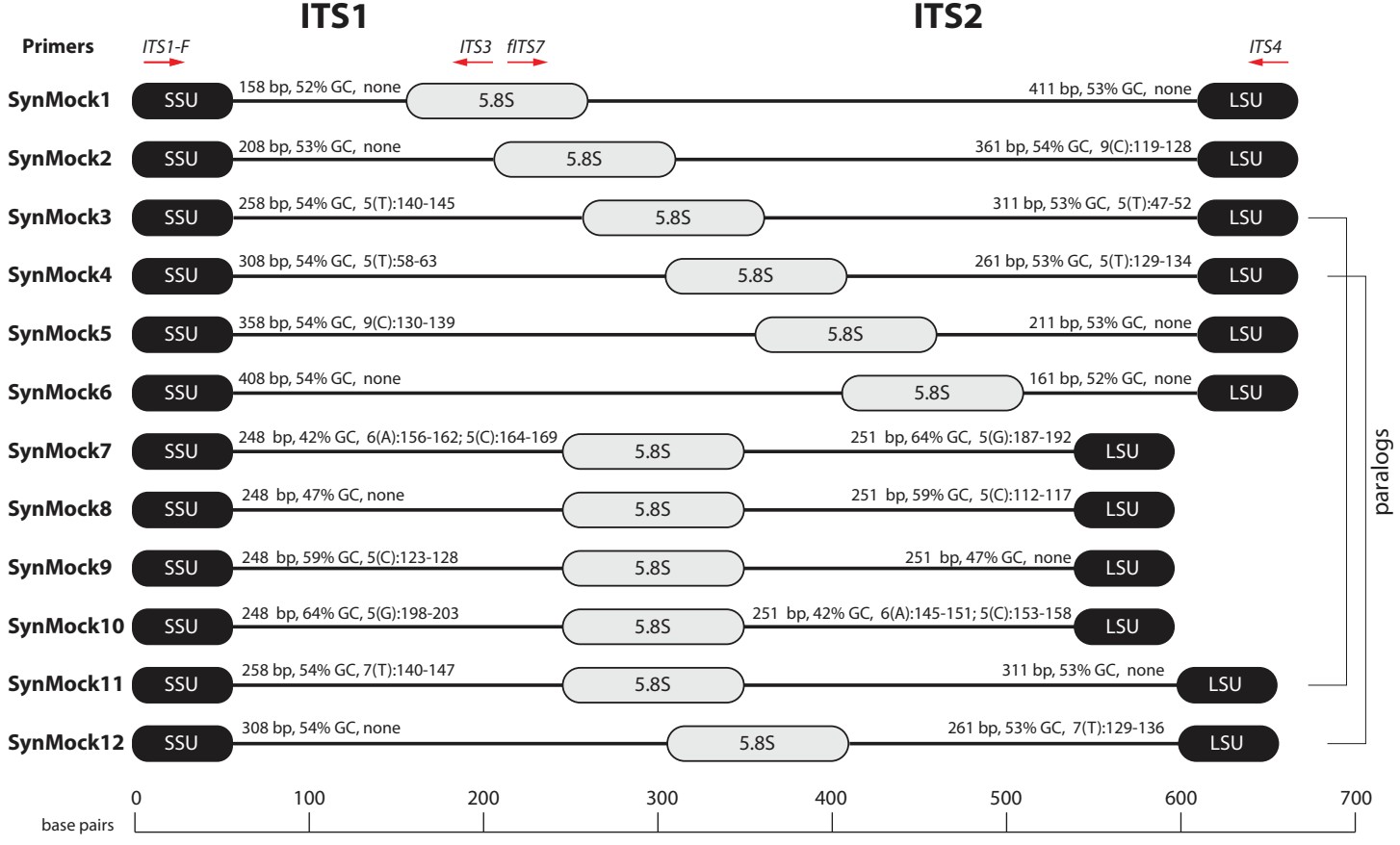

**Figure 5 Schematic drawing of the 12-member non-biological synthetic mock community (SynMock).** Conserved priming sites for either ITS1 or ITS2 amplicons are retained for versatility. The length distribution, GC content, and homopolymer stretches are representative of curated public databases; however, the sequences are non-biological and thus not found in nature.

Illumina libraries that had shown high tag-switching on the first MiSeq flowcell that did not contain the BioMock (Fig. S3). One problem that we noticed in measuring tag-switching using a mock community of actual ITS sequences (BioMock) was that these same taxa in the mock community could be present in environmental samples, which would lead to inaccurate estimation of tag-switching. In our environmental data, it was likely that at least one of the BioMock members was present in several of the environmental samples, suggesting the calculated tag-switching could be over-estimated. To overcome this problem, we designed a non-biological SynMock community composed of ITS-like sequences that contained conserved priming sites (SSU and LSU regions), ITS1 region, 5.8S region, and an ITS2 region (Fig. 5). We designed the ITS1 and ITS2 portions of the sequences to be non-biological; that is, no similar sequences are known to occur in nature (based on searches of known databases and based on the infinitesimally low probability that a randomly generated sequence would match something found in nature) and therefore these non-biological sequences can be used to accurately track tag-switching in HTAS studies. Using the summary statistics from the analysis of the UNITE reference database for guidance, we also varied the length, GC content, and homopolymer stretches to be representative of real fungal ITS sequences.

The SynMock was tested as a spike-in control on both the Ion Torrent and Illumina MiSeq platforms. The raw data were processed using AMPtk and clustered using UPARSE. These data illustrate that the synthetic sequences are able to be processed simultaneously with real ITS sequences and provide a way to track the level of tag-switching between multiplexed samples (Fig. 6). The increased benefit of being able to track the SynMock sequences as they "bleed" out of the sample allows for a more accurate measurement of tag-switching. Using default Illumina de-multiplexing (allowing one mismatch in the index sequence), tag-switching using the SynMock community was 0.057% (Fig. 6C). To determine if allowing mismatches in the index reads was increasing tag-switching, we reprocessed the data with 0 mismatches and found that tag-switching was reduced to 0.036%. While tag-switching was reduced by nearly half, the tradeoff was that 0 mismatches resulted in approximately 10% fewer reads. For most datasets, a loss of 10% of the sequencing reads should not be problematic, especially if the benefit is to reduce sample mis-assignment in the data. We noted that in our Illumina dual-indexing library prep that there was increased tag-switching on samples that had a shared reverse index (i7), suggesting that errors are increased at later stages of an Illumina sequencing run (Fig. 6B). A similar pattern of increased tag-switching correlating with relaxed primer mismatch settings was observed with Ion Torrent PGM data, although not as drastic. Allowing one mismatch in the barcode resulted in 0.167% tag-switching while allowing 0 mismatches in the barcode resulted in 0.156% tag-switching (Fig. 6C). While these data would suggest that tag-switching is perhaps higher in Ion Torrent PGM datasets, we have subsequently used the SynMock on more than 10 different HTAS Ion Torrent PGM experiments and have since seen much lower levels of tag-switching, occasionally approaching zero.

Many environmental samples can contain hundreds of taxa and thus a legitimate concern is that the 12-member SynMock community does not represent a realistic community in terms of diversity in a sample. To test if the SynMock was able to be recovered in a more complex community, we mixed SynMock together with two environmental samples that had more than 200 OTUs in previous sequencing runs. These mixed samples show that SynMock could be recovered from a complex community and the sequences behave like real ITS sequences (Fig. 6A). While many studies have set a read count threshold to filter "noisy" data from OTU tables, this threshold has been typically selected arbitrarily, i.e., OTUs with read counts less than 100 or less than 10% of the total, etc. Use of the SynMock spike-in control allowed for data-driven thresholds to be measured and moreover for the ability to filter the OTU table based on the calculated tag-switching. The AMPtk filter command calculates tag-switching by mapping the OTUs to the mock community and then provides a way to filter the OTU table based on this calculated value. AMPtk filters across each OTU in the table such that if a read abundance value for a particular sample is below the calculated tag-switching threshold it is converted to zero, and therefore difficult to sequence or "low abundance" OTUs are not indiscriminately dropped. Taken together, these data illustrate the utility of a non-biological SynMock community in parameterizing data processing steps and importantly providing a method in AMPtk to reduce tag-switching from HTAS datasets. AMPtk

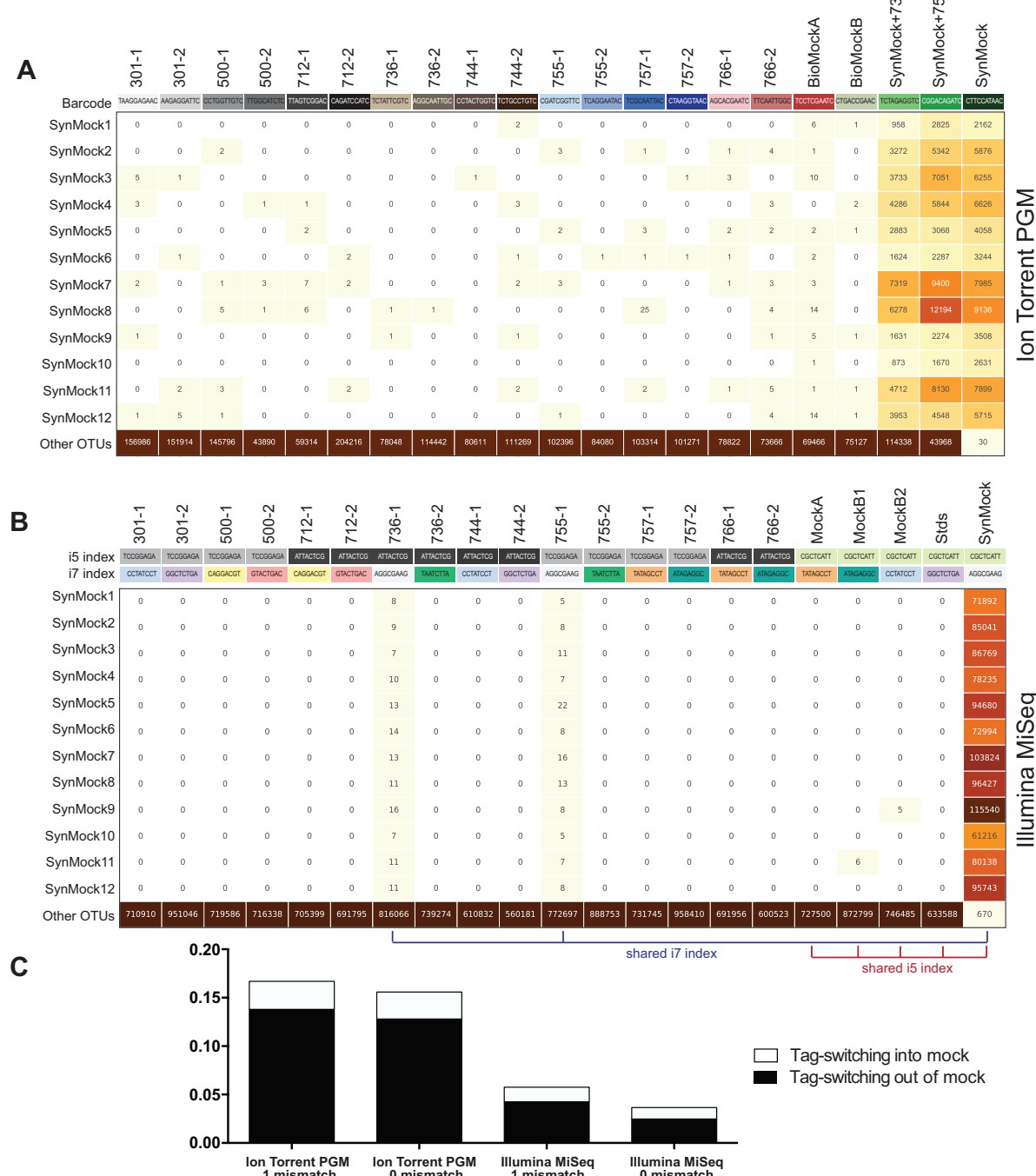

**Figure 6 Tag-switching or sample mis-assignment occurs on both Ion Torrent PGM and Illumina Miseq.** (A) Read counts from the SynMock community run on the Ion Torrent PGM platform. SynMock reads can be found in environmental samples and reads from the environmental samples are found in the SynMock sample. The data were processed allowing zero mismatches in the barcode sequence and there is no clear pattern to tag-switching on the Ion Torrent PGM platform. (B) Data processed on the Illumina MiSeq (2 × 300) allowing zero mismatches in the index reads show tag-switching in and out of the SynMock sample. Samples that share an index (i5 or i7) show an increase in tag-switching. (C) Tag-switching between samples can be tracked using the SynMock spike-in control, where AMPtk will measure both tag-switching into the SynMock as well as tag-switching into other samples. These calculated values are then used by AMPtk to filter an OTU table to remove read counts that fall below the tag-switching threshold. Tag-switching is reduced if zero mismatches are allowed in the barcode/index sequence; however, this is still not sufficient to eliminate tag-switching.

provides an easy to use method to accurately process variable length amplicons, cluster them into OTUs or denoise sequences, generate an OTU table, filter the OTU table for tag-switching, and assign taxonomy.

## DISCUSSION

Many HTAS studies have the goal of measuring and comparing biological diversity in environmental samples; however, there are technical limitations that need to be understood in order to reach justifiable conclusions. Mock communities and negative controls have been shown to have great utility for HTAS studies, and expanding upon this concept, we present a non-biological SynMock community of ITS-like sequences for use as a technical spike-in control for fungal biodiversity studies. Additionally, we describe AMPtk, a software tool kit for analyzing variable length amplicons such as the fungal ITS1 or ITS2 molecular barcodes. These two tools can be coupled together to validate data processing pipelines and reduce tag-switching from OTU tables prior to downstream community ecology analyses. The concept of a non-biological synthetic spike-in control can be expanded to many different genes and organisms, as was recently described for 16S for microbiome studies (*Tourlousse et al., 2017*).

The ITS region is widely used as a molecular barcode in fungal biodiversity studies as it is easy to amplify and public reference databases are robust. However, HTAS with the ITS region presents some unique challenges due to variability in sequence characteristics such as length and copy number. Most HTAS software development and optimization has been focused on the 16S molecular barcode, a region that is near uniform in length across prokaryotic taxa. Thus, there is a need for a software solution that can more accurately account for variable length amplicons. We developed a single-copy mock community based on cloned ITS sequences as a tool to validate and compare different NGS platforms and data processing pipelines. Using an artificial single-copy mock community of cloned ITS sequences in plasmids (BioMock), we determined that the core clustering/denoising algorithms work for variable length amplicons; however, pre-processing techniques widely used for uniform length amplicons introduce significant error into the pipelines. Simplifying the pre-processing of sequencing reads (i.e., identifying unique sequence barcodes, removing forward/reverse primers, and trimming reads to a uniform length without data loss) resulted in large improvement in downstream OTU clustering. The pre-processing of reads prior to quality filtering is critical for variable length amplicons and is implemented in AMPtk. It should be noted that we originally developed AMPtk several years ago to address a need in pre-processing HTAS data from the Ion Torrent platform, more recent versions of USEARCH have taken a similar approach for pre-processing reads allowing for the ability to pad reads with N's. However, there is no documented method to remove reverse primers using USEARCH (v10.0.240).

Proper pre-processing of variable length amplicons improves clustering results substantially. However, the BioMock results illustrated that read abundances obtained from HTAS are not a reliable proxy for inferring biological abundance, demonstrating additional assays such as qPCR or metagenomics are required to capture biological abundance. Previous studies have tried to correct for relative abundance by normalization

to rRNA copy numbers in 16S studies with minimal success (*Edgar, 2017*; *Kembel et al., 2012*). However this is not something that is currently feasible for fungal HTAS due to lack of knowledge in rRNA copy numbers. It has also been suggested that data transformations (log, square-root, etc) can be used to reduce the effects of abundance bias prior to downstream ecological analysis (*Nguyen et al., 2015*). While the data presented here do not support using read abundance as a proxy for biological abundance in HTAS studies, it does support use of presence/absence (binary) metrics as we were able to recover all members of our mock community using AMPtk, even when they were spiked into a diverse environmental sample. We identified the initial PCR reaction (library construction) as the major source of read number bias, a conclusion consistent with the literature (*Jusino et al., 2017*; *Polz & Cavanaugh, 1998*; *Wu et al., 2010*). To reduce PCR artifacts for any assay it is generally accepted that one should use the fewest cycles possible, avoid samples with low DNA quantity, and to use a proofreading polymerase (*Oliver et al., 2015*). We have tested DNA concentration and PCR cycle numbers for HTAS library generation and subsequent sequencing on the Ion Torrent PGM platform, and our results were consistent with these general guidelines (Fig. S1). However, following these guidelines is not sufficient to reduce the bias in read abundance from a mixed community from PCR. The Ion Torrent PGM platform currently has an amplicon size limit of ~450 bp, and thus some very large ITS sequences are difficult to sequence. However, there are only a small number of known ITS1 or ITS2 sequences that are longer than 450 bp (Table 1) and therefore either platform, Ion Torrent or MiSeq, provided similar results under the conditions tested.

Tag-switching has recently been acknowledged by Illumina (https://tinyurl.com/illumina-hopping), although they limit their acknowledgement to a new flow cell on the HiSeq and NovaSeq platforms. Several studies have shown that older instruments/flow cells have also shown tag-switching, albeit at a much lower rate (*Kircher, Sawyer & Meyer, 2012*; *Wright & Vetsigian, 2016*) and tag-switching has been identified on Roche 454 (*Carlsen et al., 2012*). A dual-indexing approach has been suggested to minimize the tag-switching in datasets (*Kozich et al., 2013*), however the dual indexing strategy we used in this manuscript was not sufficient to eliminate tag-switching. Tag-switching was much higher if samples shared one of the two index sequences (Fig. 6B), suggesting that dual indexing using unique sequences for every sample should further reduce the effects of tag-switching on the Illumina platform. Here we report a low rate of tag-switching on both Ion Torrent and Illumina MiSeq platforms. While the effective rate of tag-switching is low (<0.2%), coupled with the fact that read number is not a reliable proxy of community abundance, tag-switching in datasets being analyzed by presence-absence metrics is a problematic scenario. To identify and combat tag-switching, we created a non-biological SynMock community (SynMock) of ITS-like sequences that behave like real ITS sequences during the HTAS workflow. Because the SynMock sequences are not known to occur in nature, they can be effectively used to measure tag-switching in a sequencing run.

## CONCLUSION

We propose that HTAS studies of fungal ITS communities can be improved by employing SynMock or a similar non-biological community as a technical control. Additional

controls such as a BioMock or a community of mixed fruiting bodies, spores, hyphae, etc. of taxa of interest are also useful if the experiment is designed to identify the prevalence of particular taxa. The bioinformatics pipeline presented here, AMPtk, was developed to specifically address the quality issues that we have identified by using spike-in mock communities and to provide the scientific community with a necessary tool to study fungal community diversity. AMPtk is a flexible solution that can be used to study other regions used in HTAS, such as mitochondrial cytochrome oxidase 1 (mtCO1) of insects and the LSU of the rRNA array. The goal of AMPtk is to reduce data processing to a few simple steps and to improve the output of HTAS studies. Due to the inherent properties of HTAS and the ITS molecular barcode, we take the position that studies of this nature should be used as a preliminary survey of taxa present in an ecosystem and that inferring relative abundance from read numbers should be avoided. To understand relative abundance of particular taxa in a community, sequence data from HTAS studies should be coupled with independent assays such as taxa specific qPCR or digital PCR would allow for relative abundance determinations. However as DNA sequencing continues to become more cost effective, relative abundance in fungal community ecology experiments can be obtained using metagenomics.

## ACKNOWLEDGEMENTS
We sincerely thank Rita Rentmeester for assisting with the growth of some the cultures used to create the biological mock community.

### Funding
Funding was provided by the USDA Forest Service, Northern Research Station. The funders had no role in study design, data collection and analysis, decision to publish, or preparation of the manuscript.

### Competing Interests
The authors declare that they have no competing interests.

### Author Contributions
- Jonathan M. Palmer conceived and designed the experiments, performed the experiments, analyzed the data, prepared figures and/or tables, authored or reviewed drafts of the paper, approved the final draft.
- Michelle A. Jusino conceived and designed the experiments, performed the experiments, authored or reviewed drafts of the paper, approved the final draft.
- Mark T. Banik conceived and designed the experiments, performed the experiments, authored or reviewed drafts of the paper, approved the final draft.
- Daniel L. Lindner conceived and designed the experiments, contributed reagents/materials/analysis tools, authored or reviewed drafts of the paper, approved the final draft.

## DNA Deposition

The following information was supplied regarding the deposition of DNA sequences:

Raw sequencing reads and data processing scripts are available at the Open Science Framework: Palmer, Jonathan. 2018. "AMPtk." Open Science Framework. https://osf.io/4xd9r. Sequencing data is also available at NCBI Small Read Archive under the SRP144513 accession under the BioProject PRJNA305924.

## Supplemental Information

Supplemental information for this article can be found online at http://dx.doi.org/10.7717/peerj.4925#supplemental-information.

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
