# Peer review of "Non-biological synthetic spike-in controls and the AMPtk software pipeline improve mycobiome data"

_PeerJ, doi:10.7717/peerj.4925_

## Round 0.1 · original submission · Minor Revisions

The three reviewers were equally enthusiastic about the utility, novelty and potential of this mock community and analysis platform. All reviewers also were positive about the manuscript presentation. You will see from their comments (and reviewer #3's annotated MS) a number of minor revisions that should improve the clarity, interpretation and context of this paper. I look forward to reading the revised manuscript. Thanks for publishing with PeerJ.

·

Basic reporting

no comment

Experimental design

no comment

Validity of the findings

no comment

Additional comments

The authors propose a standard method for internal control of metabarcoding data and related bioinformatics workflow. The techniques seem to be well refined and useable; minor comments are given below.

96 what do you mean by ’spike-in’? I have seen that ‘spiking’ is ascribed to the procedure, where a known-quantity control DNA/organism is co-extracted and co-amplified with a sample to provide information about the sample DNA quantity / extraction efficiency. Give a clear definition. Would your amplicons also work when adding a known quantity to a sample to be subjected to DNA extraction?
99 mock communities alone so not provide information about extraction success from soil or wood.
113 ‘index bleed’ seems to have the most ambiguous meaning of the synonymous terms. Why prefer this one?
134 ‘range of fungi, including paralogs,’ What is a paralogous fungus? Do you also mean species =fungi here?
280 what about ITS3 priming site? As ITS2 site?
374 I think that most authors consider also presence/absence but report quantitative analyses. Binary HTS data are more noisy because of tag switching and yields lower R2-values because of loss of quantitative information.
382 while this is true for absolute terms, you cannot claim this for relative term. It should be tested by varying the relative abundance of ingredient taxa and running regressions.
384 it would be really interesting to test this by running the entire mock using a metagenomics approach with no PCR.
407 see also a review of Balint et al. (2016) how information about sequencing depth can be included. Discussion of rarefying may be here beyond the scope.
472 give a formula, how this is calculated for each OTU. The main problem is with OTUs that dominate in some samples but are rare in others.
509 here you are mixing relative and absolute abundance.
516 highly concentrated DNA produce chimeras – a lot!
522 For IT, it is actually quality in the sequence end that becomes critical.
* The AMPtk platform is missing completely from the Discussion. At minimum, you should compare its features with other platforms that offer several analysis options, e.g. QIIME, PipeCraft, etc.

Leho Tedersoo,
Non-anonymous referee

·

Basic reporting

see comments below - this section "passes".

Experimental design

see comments below - this section "passes".

Validity of the findings

see comments below - this section "passes".

Additional comments

This study focuses on how mock communities can be used to validate and improve the results from high throughput amplicon sequencing (HTAS) of fungi. This tool is very widely used in microbial community profiling currently and efforts to check the biological accuracy of the results are continually needed. The authors combine data based on two types of mock communities to show that there are important shortcomings with current HTAS analysis pipelines and then present a new pipeline (amptk) that overcomes many of those issues. While I would usually worry that blending the introduction of a new analysis pipeline with results from multiple mock validations would result in a very challenging paper to read, the authors do an excellent job of integrating the two in a way that is synergistic. Overall, the basic reporting is well articulated and the experimental tests are both comprehensive and easy to understand. Given the range of tests presented, the findings appear to be valid and well justified. I think this paper and pipeline represent an excellent addition to the field of HTAS-based studies of fungi and look forward to applying the Symmock and amptk pipeline in my own studies. Below are a range of relatively minor comments that I hope will help the authors improve the final presentation of their work.

L183- What is the justification for the higher then lower annealing steps in the PCR reaction – trying to reduce non-template amplification?

There are many parts of the results section that are redundant with the methods section. For example on L 305-307 there is a re-explanation of the cultures chosen for the Biomock. Some additional detail is included, but I think that that info should have been presented earlier and then this paragraph could be eliminated from the results, since it is about “creating” the mock rather than explaining the results from that mock (which is what should be in the results section). This may be a stylistic choice by the authors, but I think that making the methods a bit longer and the results more focused on the results will allow the reader to more clearly follow the manuscript.

L371-373 – Ihrmark et al. 2012 Figure 4 also shows a strong relationship between relative abundance of amplicons and relative abundance of template.

L461-462 – This is a really important additional test and I am excited to see the authors test the effect of sample OTU richness on recovery of the SynMock. It looks like despite its simplicity in OTU richness, the SynMock is capturable in more rich samples and therefore likely a good representative mock to use.

L508-513 – Switching from relative abundance to incidence based analyses may have some major downstream consequences for ecological interpretation, so I suggest the authors consider also acknowledging that data transformations of abundance based data may help account for the skew present in HTAS datasets without the massive up-weighting of rare taxa that happens with presence-absence transformations.

L518-520 – This is similar to above comment about the structure of the manuscript. Here the authors say that they tested the effect of cycle number (which they do), but I could not find the formal presentation of that test in the results section. It would make more sense to me to have a clear set of sentences describing the result in the results section than having the reader go look in the supplemental figures to see the results.

L553 – I don’t disagree with this conclusion, but I am afraid the “horse is already out of the barn” on the use of abundance-based data in HTAS datasets for fungi. The Taylor et al. (2016) paper makes a strong conclusion that read abundance is valid to use as a proxy, although their biomock is small (but phylogenetically broad). One possibility is for the authors to run their own biomock with the Taylor et al. primers and see if they get better results than the ITS2 primer pair they used. I’ll be surprised if they do and actually strongly agree with the authors that initial PCR-stage stochasticity is likely a major driver of variation in sequence reads in HTAS datasets (based on similar analyses in my lab). Unfortunately, that is not an easily fixable problem as natural samples have many taxa that can’t be amplified individually and then pooled after PCR. That said, Egan et al. (in press at Fungal Ecology) clearly shows that the repeatability of HTAS read abundances of replicates for a given sample is very high on Illumina. I think that some discussion by the authors reconciling the high precision across replicates with the high stochasticity of initial PCR would be helpful in advising readers on how to proceed with interpreting HTAS-based data.

Non-anonymous review by Peter Kennedy

Reviewer 3 ·

Basic reporting

Paper reads well. Need to cite more literature. Please see below.

Experimental design

Experimental design is appropriate, however, the authors need to make sure that their designs are not out of date. Please see below for more details.

Parameters for processing data is requested.

Validity of the findings

Discussion should incorporate a few more current literature.

Additional comments

In "Non-biological synthetic spike-in controls and the AMPtk software pipeline improve mycobiome data" the authors created biological and synthetic mock communties to test the quality of sequence data processing through a pipeline that they have developed. The authors took some of the most important needs in the amplicon processing and solved them through this pipeline. This is important to the overall scientific community. However, because the field moved so quickly, some of those needs might have already been addressed by other researchers. Overall, the paper flows well, the data presented is straight-forward, and some components of the paper are crucial to the current use of high throughput amplicon sequencing.

I feel that the most important contributions of this paper is the development of the synthetic mock community, as well as the ability to use them to solve the issue of index hoping. These two components should be highlighted in the paper because it seems that more people are moving to the dual indexing system. This would be a major contribution to the scientific community.

Nature of the "story telling" narrative is appreciated and makes reading smooth and easy, however, parts of the "results" section really should be in the methods.

The problem with length in ITS might have already been solved by Robert Edgar. Please check with him and his website to see if this is the case. If the problem had been solved, then the manuscript needs to be updated to reflect that.

Please provide parameter files for the UPARSE pipeline when used.

The results here showing that equimolar mock communities do not so equal number of sequences as an output has been found in many other papers. I would suggest that the authors cite some of them.

The discussion of whether relative abundance or presence/absence should is still in debate among the community, but the authors should at least acknowledge that over a dozen papers have attempted to correct for relative abundance. I would highly suggest that they be cited and discussed.

Annotated reviews are not available for download in order to protect the identity of reviewers who chose to remain anonymous.

---

## Round 0.2 · accepted · Accept

Thanks for your conscientious edits. I look forward to seeing your MS in print.

#